# Advancements in Artificial Intelligence-Enhanced Imaging Diagnostics for the Management of Liver Disease—Applications and Challenges in Personalized Care

**DOI:** 10.3390/bioengineering11121243

**Published:** 2024-12-09

**Authors:** Naoshi Nishida

**Affiliations:** Department of Gastroenterology and Hepatology, Faculty of Medicine, Kindai University, 377-2 Ohno-Higashi, Osakasayama 589-8511, Japan; naoshi@med.kindai.ac.jp; Tel.: +81-72-366-0221

**Keywords:** artificial intelligence, deep learning, liver disease, hepatocellular carcinoma, imagining, diagnosis, treatment, personalized medicine

## Abstract

Liver disease can significantly impact life expectancy, making early diagnosis and therapeutic intervention critical challenges in medical care. Imaging diagnostics play a crucial role in diagnosing and managing liver diseases. Recently, the application of artificial intelligence (AI) in medical imaging analysis has become indispensable in healthcare. AI, trained on vast datasets of medical images, has sometimes demonstrated diagnostic accuracy that surpasses that of human experts. AI-assisted imaging diagnostics are expected to contribute significantly to the standardization of diagnostic quality. Furthermore, AI has the potential to identify image features that are imperceptible to humans, thereby playing an essential role in clinical decision-making. This capability enables physicians to make more accurate diagnoses and develop effective treatment strategies, ultimately improving patient outcomes. Additionally, AI is anticipated to become a powerful tool in personalized medicine. By integrating individual patient imaging data with clinical information, AI can propose optimal plans for treatment, making it an essential component in the provision of the most appropriate care for each patient. Current reports highlight the advantages of AI in managing liver diseases. As AI technology continues to evolve, it is expected to advance personalized diagnostics and treatments and contribute to overall improvements in healthcare quality.

## 1. Introduction

Liver diseases often lack specific symptoms and tend to progress chronically, frequently leading to serious conditions such as liver cirrhosis and hepatocellular carcinoma (HCC), which make them a significant public health issue. In recent years, advancements in pharmacological therapies have enabled the effective control of hepatitis B and virus (HBV) and hepatitis C virus (HCV) [1]. However, the rising incidence of non-viral liver disease, such as MASLD (metabolic-associated steatotic liver disease) and MASH (metabolic-associated steatohepatitis), has contributed to an increase in patients with cirrhosis and liver cancer [2]. Therefore, proper management of liver diseases has become a critical challenge.

On the other hand, imaging modalities such as ultrasound (US), computed tomography (CT), and magnetic resonance imaging (MRI), are commonly employed and are indispensable in the diagnosis and treatment of liver diseases, with a large volume of medical imaging data accumulating daily. However, medical doctors are at risk of missing lesions or making misdiagnoses in clinical practice because of time constraints and fatigue. These risks are particularly pronounced for beginners or non-specialists, for whom interpreting complex medical images is sometimes a significant challenge. Artificial intelligence (AI)-aided diagnosis can be a promising solution to mitigate such human errors. Since most medical images are standardized in the Digital Imaging and Communications in Medicine (DICOM) format, they provide a substantial advantage as training data for AI. AI, trained on lesion-specific imaging features, can offer valuable information not only for disease diagnosis but also to estimate disease progression and predict the prognosis, thereby enhancing disease management. Currently, this area has garnered substantial research attention, with numerous studies focused on the development and validation of AI-driven approaches to improve the reliability of liver disease diagnostics [3].

This paper outlines the current state of AI development in imaging diagnostics for the management of liver diseases and its societal implementation, primarily focusing on deep learning models. In particular, the importance of AI-aided diagnosis in imaging is discussed in the context of its applications and challenges in personalized care.

## 2. Advantages of AI in Medical Imaging Diagnostics

Liver diseases, such as cirrhosis and malignant liver tumors, significantly impact a patients’ prognoses, making early diagnosis and timely therapeutic intervention crucial. Imaging diagnosis is generally non-invasive and vital to diagnosing and managing liver diseases (e.g., screening for early diagnosis among a high-risk population and selection of the most suitable treatment). However, the large volume of imaging data often overwhelms the clinical capacity, making it challenging to review and interpret results within a limited time frame. AI, trained on extensive image datasets labeled by specialists, has demonstrated diagnostic accuracy that, in some cases, surpasses that of human experts. This enhances the standardization of imaging diagnostic quality [4,5,6,7,8]. Moreover, AI systems can process large amounts of data rapidly, significantly reducing the time required for diagnosis.

Currently, AI systems trained with medical images as input data have been shown to achieve diagnostic accuracy comparable to that of specialists and, in some domains, even surpass them. More importantly, several AI systems have been reported to perform tasks that are considered extremely difficult for humans. For instance, an AI technology utilizing deep learning with a convolutional neural network (CNN) was reported to successfully perform automated detection of diabetic retinopathy and diabetic macular edema in retinal fundus photographs [9]. Another AI system was reported to be able to determine the presence or absence of lymph node metastasis in breast cancer from pathological specimens, demonstrating performance that surpassed diagnoses made by specialists over the course of several hours. This AI system was capable of providing diagnoses within seconds, showcasing its exceptional efficiency [10]. Additionally, there have been reports that AI has achieved diagnostic accuracy equivalent to that of dermatologists in identifying malignant skin tumors [11]. Further studies have documented the ability of AI to analyze optical coherence tomography images to distinguish critical conditions such as age-related macular degeneration and diabetic macular edema [12]. An AI model trained on 32,537 whole-slide images (WSIs) of hematoxylin-eosin (HE) stained specimens has been developed to predict the origin of tumors, extracting features to diagnose cancers of unknown primary origin [13]. The ability of AI to identify image features that are unrecognizable to humans proves its potential to enable earlier therapeutic interventions and play a crucial role in clinical decision-making [14,15].

## 3. Training of AI Models Using Imaging Data of Liver Disease

The common imaging modalities for liver diseases include US, CT, MRI, and pathological examination [16]. Although these modalities are frequently used in the management of liver disease, there are differences in the characteristics of the images that relate to their application as AI training data.

US examinations provide real-time images and are often used for the initial diagnosis of liver lesions. If a lesion is suspected, further examinations such as CT or MRI are typically conducted. Given the widespread availability of diagnostic US equipment, a large number of images can be obtained for AI training. However, during imaging, US waves are directed at the target object and the reflected echoes are processed into the image data, making issues such as artifacts common. Therefore, quality control of the training images is crucial for effective AI development [5,17,18]. In contrast, CT and MRI scans offer high spatial resolution and allow for comprehensive imaging of the liver, enabling a three-dimensional reconstruction of its structure. These modalities provide images with a more uniform quality, making radiomics data more readily available compared to US images. Additionally, in liver CT or MRI imaging, classification of the liver segment is readily achieved by analyzing the anatomical pathways of the hepatic and portal veins [5]. Nonetheless, caution is required when using such images for training due to potential issues such as hepatic deformation due to liver atrophy caused by cirrhosis or partial surgical resection.

Pathological examination remains the gold standard for diagnosing many liver diseases. The advent of digital pathology scanners has made it easier to obtain WSIs for AI training. However, issues regarding image quality due to sample degradation may arise when using stored specimens from older cases. While WSIs are often divided into patches for AI training, annotating images at the cellular level is labor-intensive. Therefore, careful consideration of the intended purpose and functionality of an AI system is essential to ensure appropriate annotation.

### 3.1. Supervised Learning for Imaging Data

In supervised learning, AI is trained using expert-labeled data provided by specialists to match its final output with the correct labels. In traditional methods, specific features such as the shape, extent, and contrast of lesions need to be quantified for the training of algorithms like support vector machines, logistic regression, and multilayer perceptrons for classification and prediction [19,20]. When humans manually select features, the results are often easier to interpret. In contrast, deep learning automatically determines numerous features while simultaneously training the model using multilayered neural networks. Each layer of the network extracts features from the previous layer, transforming them progressively until the final output aligns with the correct label. CNNs are commonly used in image analysis [5]. CNNs use specific coefficients, or kernels, to process pixels within a defined range, thereby extracting diverse image features (Figure 1a). By automatically adjusting the kernel coefficients and sizes, CNNs can capture various image features. This approach consolidates information within a set range of pixels, reducing the influence of minute positional relationships and ensuring stable, unbiased learning. Some models trained on time-series data from different stages of the same case have been applied in order to distinguish malignant liver tumors. However, preparing a large number of expert-level labels for the training data is often challenging due to the significant time, effort, and cost required for annotation. To address this, innovative approaches such as collectively labeling datasets to assess the overall disease conditions are necessary, particularly for complex annotations such as patch images of WSIs in pathological tissue [6,21].

### 3.2. Unsupervised Learning for Imaging Data

Unsupervised learning aims to detect specific patterns within the data, without the need for labeled training data. This approach includes techniques such as clustering, which groups subjects based on data features, and dimensionality reduction, which extracts key information that characterizes the data. One advantage of unsupervised learning is that it requires less effort to prepare the training data, since no labels are required. However, even if new patterns are discovered, interpreting such classifications can be challenging and AI-generated classifications may not always be clinically useful. Since unsupervised learning lacks labels, it cannot perform differential diagnoses or predict diagnostic probabilities, limiting its application in clinical practice. Clustering techniques such as k-means and spectral clustering are used to classify liver MRI and CT images based on the distances derived from the sample features [22]. Other applications include autoencoders, which transform images into vectors that retain important information while removing irrelevant details. Generative adversarial networks (GANs) are also employed, whereby a generator creates artificial images and a discriminator attempts to differentiate between the real and generated images (Figure 1b). This technique can be used to generate new artificial data, which may serve as training data for AI [4,23].

### 3.3. Transfer Learning for Imaging Data

Transfer learning is commonly employed to address the lack of labeled training data [24]. It involves using a pre-trained AI model, initially developed for a specific purpose, as a starting model for performing similar tasks (Figure 1c). This method simplifies the construction of models compared to training an AI model from scratch. For instance, there have been reports of a pre-trained AI model being used for contrast-enhanced US images to develop an AI for diagnosing lesions in B-mode US using a smaller dataset [25]. Additionally, transfer learning has been applied to develop models for classifying liver fibrosis by using a pre-trained AI model on B-mode US images [26].

## 4. AI-Aided Imaging Diagnosis and Its Clinical Application

AI plays a significant role in clinical practice, aiding in the diagnosis and risk prediction of diseases and serving as a powerful tool for personalized medicine [8,27]. However, developing versatile AI systems requires the collection of large-scale medical data. Many reports on AI in medical imaging have issues such as small and homogenous training cohorts and a lack of validation with external datasets [18]. Despite these challenges, AI is expected to contribute to various aspects of disease management, including the early detection of lesions, accurate differentiation of malignant tumors, and the prediction of disease progression and complications. AI not only supports current medical practices by reducing human error and variability among clinicians, but also has the potential to identify features in images that are undetectable to humans and apply these features as morphological biomarkers in clinical settings.

AI can be useful in several clinical setting such as ① disease prevention, ② early detection of lesions, and ③ the selection of effective treatments (Table 1). For instance, lifestyle improvements such as reducing alcohol consumption can prevent the onset of certain diseases and reduce deaths related to alcohol use. AI-based applications that encourage behavior changes, such as reducing alcohol intake, could be effective tools for disease prevention. Early detection is another key factor in improving prognosis. AI can be applied to identifying populations at high risk of diseases and developing new screening methods. A recent AI model, trained on a large dataset from Denmark and the USA, successfully predicted the occurrence of pancreatic cancer within 36 months [28]. Since pancreatic cancer is difficult to detect early and has a poor prognosis, an AI system that identifies high-risk individuals could greatly contribute to early detection and improved outcomes. For example, many AI systems have been reported to narrow down groups at high risk of HCC effectively, such as those that estimate the stage of liver fibrosis and steatosis. Alternatively, AI systems that assist in image diagnosis are also expected to be effective in reducing misdiagnosis and aiding early detection. AI has been shown to support the early diagnosis of pancreatic cancer from plain CT images [29]. In liver disease, AI models have been reported to support the diagnosis of liver tumors from US and CT images, and these systems are expected to reduce human error and improve diagnostic accuracy. Selecting effective treatments is also crucial for improving disease prognosis, and AI models that estimate treatment outcomes and prognoses have also been reported. For example, an AI system using image-based diagnostics has been reported to estimate and compare survival outcomes between surgery and radiofrequency ablation (RFA) in patients with early-stage HCC [30].

For the prevention of disease, lifestyle improvements are necessary and AI application may help promote behavior and lifestyle change. For the diagnosis of disease, AI can support disease screening and diagnosis to narrow down high-risk populations and examine them with high accuracy. For treatment, the prediction of treatment outcomes by AI should be helpful for the selection of the most effective treatments for each patient.

### 4.1. AI-Assisted Imaging in the Diagnosis of Hepatic Steatosis and Fibrosis

Progression in liver diseases is strongly associated with hepatic steatosis and fibrosis, conditions that elevate the risk of cirrhosis and HCC, especially in cases of MASLD which was previously referred to as non-alcoholic fatty liver disease (NAFLD). Timely diagnosis of these conditions is critical. Although traditionally reliant on biopsy for diagnosis, this risks inaccurate assessment due to the limited sample volume which may fail to accurately capture the extent of steatosis and fibrosis. Moreover, biopsies are invasive, which complicates repeated assessments. Therefore, recent efforts have focused on developing non-invasive, more reliable predictive AI models through the training of AI with a large set of imaging data to complement or replace biopsies.

#### 4.1.1. AI Models for Staging and Early Diagnosis of Liver Fibrosis

Liver fibrosis and cirrhosis are well-established high-risk factors for developing liver cancer, including HCC and intrahepatic cholongiocellular carcinoma (ICC). Non-invasive, accurate assessment of liver fibrosis is crucial not only for the early diagnosis of cancer, but also for managing life-threatening complications. So far, many AI models have been reported for staging and early diagnosis of liver fibrosis. For example, Ai et al. proposed a deep learning model based on US imaging to evaluate liver fibrosis. This model employs radio-frequency signals from automatically segmented regions in the liver for evaluation in B-mode US images, reducing the reliance on manual input while increasing the accuracy. The model achieved high area under the curve (AUC) scores of 0.957 for ≥F1 fibrosis and 0.808 for ≥F2 fibrosis in the receiver operating characteristic curve (ROC), demonstrating its effectiveness, especially in classifying early-stage liver fibrosis [31].

Park et al. and Liu et al. also demonstrated the effectiveness of AI in the staging of liver fibrosis according to the METAVIR scoring system using US images [32,33]. These studies included multiple deep convolutional network (DCNN) models, including VGGNet, ResNet, DenseNet, EfficientNet, and ViT, with AUCs ranging from 0.95 to 0.96. EfficientNet demonstrated the highest sensitivity, achieving an average sensitivity of 0.85. These findings suggest that DCNN-based models can classify fibrosis stages with high-accuracy, even in settings without intervention by an experienced radiologist. Another report on AI-assisted non-invasive methods of evaluating liver fibrosis by analyzing images from various modalities, including US, CT, and MRI, revealed its ability to achieve diagnostic accuracy comparable to that of human experts [34].

In contrast, some studies applied multimodal data to develop an AI system for the evaluation of liver fibrosis. Zha et al. developed a combined model of radiomics and clinical data (CoRC) to improve the diagnostic accuracy for liver fibrosis (≥F2) through automated liver segmentation and deep learning analysis of MRI images. This model demonstrated superior accuracy over conventional fibrosis scoring methods like fibrosis-4 (FIB-4) and aspartate aminotransferase to platelet ratio index (APRI), achieving AUC values of 0.79–0.82 on the testing datasets. When combined with the transient elastography liver stiffness measurement (TE-LSM), CoRC provided enhanced diagnostic precision, suggesting promising applications in clinical practice [35]. Chen et al. also developed a sequential algorithm that combines US image-based deep learning models with conventional non-invasive scores for the assessment of fibrosis, including FIB-4 and the measurement of shear wave elastography (SWE) [36]. This approach, employing FIB-4, FIB-Net (a US-based DL model), and SWE in sequence, improves diagnostic accuracy in advanced fibrosis cases. The algorithm demonstrated a specificity of 94% and a positive predictive value (PPV) of 73%, surpassing the two-step method of FIB-4 and SWE in distinguishing cases that need further examination, both of which are recommended in the guidelines of the European Association for the Study of the Liver (EASL) [37].

#### 4.1.2. AI Models for Staging and Early Diagnosis of Steatotic Liver Disease

Reports analyzing the performance of AI-aided diagnostic models for NAFLD (or MASLD) based on US imaging (including elastography) and clinical (serological) data showed that these models not only provide efficient diagnostic support through AI, but also contribute to the reduction of treatment costs as a result of early diagnosis, positioning them as potential alternatives to invasive liver biopsies [38].

More specifically, Wang et al. developed a deep learning model based on SWE imaging, which visualized tissue elasticity and accurately staged fibrosis in a multicenter study [39]. Santro et al. evaluated the impact of AI algorithms on US imaging for diagnosing steatotic liver disease (SLD) [40]. Conventional US diagnosis often relies on the hepatorenal index, assessing contrast differences between the liver and kidney. However, variability among operators remains a challenge. In this study, the AI algorithm calculated the hepatorenal index automatically, showing consistent diagnostic accuracy in differentiating mild from moderate-to-severe SLD with an AUC of 0.98. Njei et al. also reported on a machine learning model using Extreme Gradient Boosting (XGBoost) for the early identification of patients with MASH. This model integrates factors including liver stiffness measured through US elastography, achieving high diagnostic accuracy with an AUC of 0.95, sensitivity of 0.82, and specificity of 0.91. Moreover, the model utilizes the impact of predictors on high-risk MASLD prediction using the SHapley Additive exPlanations (SHAP) values, enhancing its clinical usability by providing clinicians with understandable results. This model demonstrates superior predictive performance compared to traditional indicators like FIB-4, the Aspartate Aminotransferase to Platelet Ratio Index (APRI), BMI-AAR-T2DM (BARD) scores, and NAFLD Fibrosis (NAF) scores, showing promising results for clinical application [41].

Multiple meta-analyses have further substantiated the utility of AI-assisted imaging techniques in diagnosing and monitoring SLD progression. For example, it has been reported that the effectiveness of AI-supported US diagnostics in non-alcoholic SLD was shown to have a sensitivity of 0.97, specificity of 0.98, and an AUC of 0.98 for diagnosis [42]. Additionally, AI-assisted US diagnostics outperformed models that rely solely on clinical data, confirming the utility of non-invasive imaging with AI integration for stratifying at-risk groups. Kwon et al. and Zhao et al. assessed the diagnostic accuracy of AI-driven imaging for SLD. They used the Quality Assessment of Diagnostic Accuracy Studies (QUADAS) AI tool for the assessment and found the performance of the AI model to have a sensitivity of 92%, specificity of 94%, and a high diagnostic odds ratio (DOR) of 182.36. Furthermore, subgroup analyses based on region, type of algorithm, and imaging modality consistently confirmed the efficacy of AI for the diagnosis of SLD [43,44]. Integrating AI into imaging modalities not only improves diagnostic accuracy but also enables faster diagnosis, especially compared to traditional pathological examination. More importantly, in settings with limited access to specialized care, AI-assisted systems can provide high-quality diagnostics to facilitate early intervention and management of SLD, which is a known risk factor for liver dysfunction and malignant tumors as well as cardiovascular disease.

AI-assisted imaging shows significant promise in non-invasive and precise diagnoses of both hepatic steatosis and fibrosis. These AI models provide powerful tools for predicting disease progression in diffuse parenchymal liver disease to support decision-making in clinical settings.

### 4.2. AI Models for Diagnosis of Liver Tumor

AI-assisted imaging is becoming a transformative tool in medical diagnostics for liver tumors, particularly for the diagnosis of HCC which is the most common primary liver malignancy. AI-based models can detect intricate patterns in imaging data that are challenging to identify through conventional means, thus providing significant support to radiologists [45]. AI applications in the diagnosis of liver tumors encompass various imaging modalities, each with unique contributions and challenges.

US serves as a non-invasive, first-line diagnostic modality that benefits from the potential of AI to standardize image interpretation and reduce the variability caused by human perception. In the clinical setting, B-mode US is frequently recommended for screening cirrhotic patients for HCC. Differentiating benign from malignant lesions is crucial for early HCC detection [46,47,48]. AI models using B-mode US for HCC diagnosis show high diagnostic accuracy; for example a deep CNN model developed by Yang et al. achieved an AUC of 0.924, significantly surpassing the diagnostic performance of radiologists with over 15 years of experience [49]. Similarly, an AI system that has been trained on over 70,950 B-mode US images to differentiate HCC, metastatic liver cancer, hemangiomas, and cysts has been reported [50]. The performance of this AI model surpassed that of human experts in the four-class discrimination and in the discrimination of benign and malignant liver tumors. Chiang et al. reported AI that can outperform human experts in detecting early HCC using a multivariate logistic regression model based on clinical features and imaging data [51]. Because US diagnosis relies on the operator’s experience, an AI-aided system is useful for the standardization of US examination quality. However, these models faced limitations due to the exclusion of certain lesion types, which might affect their generalizability.

Contrast-enhanced US (CEUS) utilizes microbubbles to enhance blood vessel visualization, making it particularly useful for detecting HCC, especially when enhanced by AI. Guo et al. demonstrated improved diagnostic accuracy for malignancy using AI trained on CEUS images [52]. Liu et al. integrated clinical data into their CEUS-based AI model, achieving an AUC of 0.957 and outperforming experienced radiologists [53]. CEUS-based models can be useful even for complicated diagnoses which deviate from the expected enhancement patterns in standard practice, such as differentiating focal nodular hyperplasia and atypical HCCs (aHCCs) [54].

CT imaging, which is widely accessible and cost-effective, also benefits from AI. Wang et al. trained a CNN model with a vast dataset of CT images from 7512 patients, achieving an AUC of 0.883 and a diagnostic accuracy that matched or exceeded radiologists’ performances [55]. AI models in CT imaging help recognize enhancement patterns and differentiate HCCs from other liver malignancies, such as ICCs. For example, Gao et al. designed a CT-based CNN that differentiated between HCC, ICC, and metastatic lesions, with an AUC of 0.937 for HCC diagnosis [56]. Other AI models were also reported for their differentiation between HCC and ICC, as well as for comparing the diagnostic power of three-phase versus four-phase CT in HCC patients [57,58]. However, issues persist, such as distinguishing between aHCCs and benign tumors due to their similar imaging characteristics, which remains a challenge even with AI assistance.

MRI, which is known for its superior soft-tissue contrast, has become indispensable in detecting small nodules. Kim et al. developed an MRI-based CNN model that achieved an AUC of 0.970 and detected small HCC lesions more effectively than junior radiologists [59]. In MRI, the utilization of hepato-specific contrast agents such as gadoxetate disodium improves sensitivity for the detection of lesions, as it provides clear demarcation of liver tumors in the hepatobiliary phase. Hamm et al. developed an AI model for the diagnosis of focal liver lesions using 494 MRIs. The model demonstrated an accuracy of 91.9% in classifying focal lesions and a sensitivity of 90% for HCC detection, which performed well even in cases that were misclassified by radiologists [60]. Hu et al. developed an AI model to differentiate HCC from ICC. This model showed an AUC of 0.760–0.790 and 73–75% accuracy, which was below that of the radiologists (81%) [61]. Therefore, developing AI models that assist imaging diagnoses need to tackle the challenge of images that poorly differentiate hypovascular HCC from mass-forming ICC, which may exhibit similar imaging features.

## 5. Prognostic Prediction and Personalized Medicine in Liver Diseases with Imaging-Based AI

Radiomics has increasingly been employed to develop AI models for the prediction of prognoses in liver diseases, particularly for advances in personalized medicine. Here, we summarize recent studies on predictive modeling for complications in liver cirrhosis and the emergence of liver cancer, highlighting their clinical applicability and challenges.

### 5.1. Applications of AI-Assisted Imaging in Management of the Complications in Liver Cirrhosis

Predicting the risk of progression in liver diseases, such as liver decompensation, malignant potential, or liver cancer prognosis, presents major challenges. Several AI models have been developed to predict the stages of fibrosis [62], predict variceal bleeding in hepatitis B-related cirrhosis [63], and detect cirrhosis and portal hypertension for the establishment of personalized treatment based on imaging [64].

So far, imaging markers such as spleen volume have been utilized to predict the risk of liver decompensation [65]. Additionally, Yu et al. developed a model that estimates the hepatic venous pressure gradient from contrast-enhanced CT images to diagnose portal hypertension [66]. Other models related to portal hypertension include those that estimate the hepatic venous pressure gradient in cirrhotic cases using CT angiography [67], models that predict portal hypertension from CT and MRI through hepatic venous pressure gradient measurements [68], and models that detect subclinical hepatic encephalopathy, a complication of cirrhosis, using functional brain MRI [69].

Regarding the management of portal hypertension, Peng et al. focused on predicting the risk of esophageal variceal bleeding (EVB) in cirrhotic patients using an AI model with radiomic features of abdominal CT images from multiple organs. The new radiomic clinical model incorporated radiomic features from the liver, spleen, and esophageal regions along with clinical risk factors, achieving a high AUC of 0.951 in the training cohort and 0.930 in the validation cohort. Notably, leveraging the multiorgan imaging features outperformed the single-organ radiomics, enhancing the EVB risk prediction [70]. Brata et al. introduced an AI algorithm to improve inter-observer variability in the grading of esophageal varices during endoscopic examinations, facilitating precise risk stratification [71].

On the other hand, Cai et al. developed a model predicting the survival rate of patients with liver cirrhosis who received post-transjugular intrahepatic portosystemic shunt (TIPS) operations [72]. This model, which incorporated factors like age, total bilirubin, and the spleen volume-to-platelet ratio, successfully predicted the 1-, 3-, and 5-year survival rates, achieving a C-index of 0.80 and AUCs of 0.828, 0.761, and 0.729, respectively. By automating organ volume measurements through deep learning on CT images, the model improved both the diagnostic efficiency and accuracy, thereby aiding in the selection of patients suitable for TIPS and treatment planning [72].

Furthermore, AI models have shown predictive capability for the progression of liver fibrosis in patients with MASLD, and those with diabetes mellitus (DM). Dai et al. developed a deep learning model integrating US elastography images and clinical data, which predicted fibrosis progression with high accuracy in a test cohort [73]. Additionally, Li et al. reported an association between diabetic retinopathy (DR) and hepatic fibrosis in DM patients, suggesting that DR could be a predictive marker for the progression of hepatic fibrosis. In their study, a significant correlation was observed between the severity of DR and fibrosis stage, with an odds ratio of 1.521 (*p* = 0.003) and an AUC of 0.72 for fibrosis prediction in DM patients. This suggests that fundus imaging characteristics might be useful for assessing the risk of progression in patients with hepatic fibrosis [74].

In liver transplantation cases with large tumor volumes, multimodal systems for the precise assessment of liver fibrosis and risk assessments that incorporate imaging data have also been reported [15]. The ability of AI to recognize subtle changes in imaging features that may not be apparent to human observers allows for more precise monitoring of treatment responses, predicting disease progression and forecasting outcomes. For instance, Taylor et al. highlighted the advantage of image-based AI over traditional histology-based staging to evaluate liver fibrosis progression, particularly in cases of non-alcoholic steatohepatitis, where fibrosis staging can be heterogeneous [75]

These studies underline the growing potential of AI-assisted imaging, not only in liver disease diagnostics but also in establishing prognostic models risk prediction. Management enabled by imaging-based AI represents a promising area for personalized medicine.

### 5.2. AI-Assisted Imaging in Diagnosis and Management of HCC

For liver cancer risk prediction, AI models have been reported that assess the risk of postoperative recurrence by combining tumor shape and radiomics features with serum markers such as the alpha-fetoprotein and albumin-bilirubin grades (Table 2) [76]. Moreover, AI systems have been developed to predict microvascular invasion, cytokeratin 19 expression associated with malignancy, and early tumor recurrence from CT and MRI images [77,78]. Feng et al. used gadolinium-ethoxybenzyl (EOB)-diethylenetriamine-enhanced MRI (EOB-MRI), combined with the pathological findings from resected specimens as training data, to estimate microvascular invasion preoperatively in resectable HCC cases, outperforming radiologists in some cases [79]. Other studies have applied random forest models to predict recurrence-free survival after surgery from EOB-MRI features in solitary HCC cases [80]. For the prediction of early recurrence, an AI model using s preoperative EOB-MRI feature including the 3 mm area outside of the tumor edge demonstrated s performance equivalent to an AI model trained with pathological findings from resected specimens [80].

Ma, Y. and colleagues developed a diagnostic model using multiphase radiomics data from CT and MRI to distinguish between HCC and non-HCC tumors. In this study, multiple machine learning algorithms, including support vector machine (SMV), K-nearest neighbor, and random forest, were applied to the radiomic data, with clinical data integrated to optimize the model’s performance [81]. The model achieved a high diagnostic accuracy of 0.824, with the SVM-based model performing best. The approach highlights the potential of radiomics data for constructing AI models to support precision in the diagnosis of liver tumors.

Further enhancing the management of HCC by training AI models on quantitative imaging data makes it possible to accurately predict individual responses to various treatments, thereby enabling appropriate therapeutic approaches for liver cancer and contributing to reductions in costs. The ability of AI to predict the efficacy of local therapies, pharmacotherapies, and radiotherapies offers significant value in personalized medicine [82]. A summary of AI-based models trained on imaging data to predict treatment outcomes and prognoses in liver cancer is shown in Table 2. Qian et al. reported an AI model specifically designed to predict the early recurrence of HCC following hepatectomy in cirrhotic patients [83]. The model combines clinical and radiomic features, achieving high predictive accuracy with an AUC of 0.844 in the training cohort and 0.790 in the validation cohort. Using the Kaplan–Meier survival analysis, the model demonstrated clinically significant discrimination in stratificating overall survival (OS), empowering clinicians to devise targeted therapeutic strategies to improve patients’ OS outcomes.

Several studies have also demonstrated the potential for AI to predict responses to transarterial chemoembolization (TACE) using CT, MRI, and US images. For example, multimodal AI models, which integrate clinical and imaging data, have been used to predict TACE outcomes [84,85,86,87,88]. CNN-based models trained on CT images and random forest models using MRI data are examples of the AI systems developed for this purpose [84,85]. Zhang et al., explored the use of AI for forecasting therapeutic responses to conventional TACE (cTACE) in HCC patients [89]. The study involved the use of pretreatment CT parameters, including arterial, portal, and arteriovenous enhancement ratios to predict responses. Among the models tested, the random forest combined model (RF-combined) achieved the highest accuracy, with an AUC of 0.800 and a net reclassification improvement of 0.8 (*p* < 0.001). Additionally, the model’s interpretability was enhanced by using the SHAP algorithm, which identifies the influential factors in the predictions, making AI-derived insights accessible to clinicians. By leveraging AI for the prediction of treatment response, unnecessary interventions may be minimized, contributing to a reduction in the patient burden [89].

Additionally, AI models that analyze contrast agent inflow dynamics in CEUS images have also been used to predict the effectiveness of TACE [90]. AI systems trained on preoperative CEUS images can predict post-operative progression-free survival for early-stage liver cancer patients undergoing RFA or surgery, allowing physicians to adjust their treatment strategies based on AI predictions, potentially leading to better tumor control [30]. Ibragimov et al. developed a CNN-based model using 3D-CT images of HCCs before radiotherapy to predict radiation-induced liver injury [91]. Similarly, Müller et al. used AI-based segmentation to calculate spleen volume from CT images and identified it as a predictive marker for survival and estimation of the risk of hepatic decompensation [92].

The treatment landscape for liver cancer is diverse, with significant improvements seen in the survival rates for advanced-stage cases. However, the therapeutic framework has become increasingly complex, with multiple treatment options available for the same stage of HCC depending on patient-specific factors [93,94]. Despite this, appropriate biomarkers to guide treatment decisions are still lacking. In medical practice, pathological diagnosis provides crucial information for the selection of treatment and the prediction of prognosis in cases of liver malignancies. To address the shortage of pathologists and to enhance the level of expertise, the development of AI models to assist in pathological diagnoses in this field is also being actively pursued. For example, Chen et al. developed a CNN (Inception V3) trained with HE-stained tissue images from the Genomic Data Commons Databases, achieving a classification accuracy comparable to that of a pathologist with five years of experience in HCCs, with an accuracy of 96.0% for benign versus malignant classifications and 89.6% for tumor differentiation [95]. The model also successfully predicted mutations in key genes such as the *CTNNB1* and *TP53* genes, indicating the utility of deep learning to assist pathologists with both classification and genetic analysis. Kiani et al. also developed a deep learning model to assist pathologists in differentiating HCC from ICC using 70 HE-stained WSIs [96]. This AI model showed accuracy rates of 88.5% and 84.2% on the internal and external validation sets, respectively. Although there were no significant differences compared to all pathologists, its performance improved in a subset excluding pathologists with unclear levels of experience. Similarly, Feng et al. created a residual CNN model to reduce manual processing in HCC classificationd, achieving an accuracy rate of 98.7% and 87.9% on the internal and external validation sets, respectively [97].

Currently, determining the optimal therapeutic approach for liver cancer remains challenging, and conducting numerous prospective clinical trials and omics analyses for biomarker development to address these uncertainties requires substantial effort [65]. However, the predictive capabilities of AI can address these challenges by estimating the outcomes of treatment and disease progression based on historical data, offering a powerful tool for optimizing clinical decision-making. In addition, genetic mutations play a key role in liver cancer prognosis, which can be applicable in the training of AI for the prediction of patient prognoses [98,99,100]. Studies in non-small cell lung cancer have shown that prognostic imaging features can be associated with genetic expression data of cancerous tissues [101]. Similarly, for liver cancer, efforts have been made to estimate specific genetic mutations, tumor immune microenvironments, and survival from images of HE specimens (Table 2) [95,102,103]. Saillard et al. utilized WSIs to create two deep learning algorithms aimed at predicting survival outcomes for HCC patients post-surgical resection [102]. Their models demonstrated superior predictive capabilities compared to traditional scoring systems, with c-indices of 0.78–0.75. Notably, the study identified specific histological features associated with poor prognosis, highlighting the importance of integrating pathological expertise into AI models. Shi et al. introduced a weakly supervised deep learning framework that leveraged prior knowledge to extract prognostic indicators from WSIs [103]. Their research established a tumor risk score (TRS) that independently predicted patient outcomes across multiple cohorts, outperforming conventional clinical staging systems. The study elucidated key pathological features linked to TRS, such as sinusoidal capillarization and inflammatory cell infiltration, thereby providing insights into the biological underpinnings of HCC progression. These studies underscore the transformative potential of AI in enhancing the precision of pathological diagnoses and prognostication in liver cancer. In the future, multimodal AI models integrating imaging with clinical, genomic, and transcriptomic data could provide even more accurate risk estimations in cases of liver cancer [7,104].

**Table 2 bioengineering-11-01243-t002:** AI trained with imaging data to predict treatment outcomes and prognosis in hepatocellular carcinoma patients.

Imaging Modality and Prediction	Number of Training Cases *	Findings	References
Ultrasonography			
HCC: Prediction of RFS after MWA	513 cases *	Two-year RFS after MWA (C-index = 0.72)	Wu, et al., 2022 [105]
HCC: Prediction of recurrences after RFA or MWA	318 cases *	Recurrence beyond two years after RFA or MWA (C-index = 0.77)	Ma, et al., 2021 [106]
HCC: Prediction of treatment response to TACE	36 cases (CEUS)	Accuracy of 86%, sensitivity and specificity of 89% and 82%	Oezdemir, et al., 2020 [88]
HCC (early stage): Prediction of RFS after RFA or surgery	214 cases for RFA *, 205 cases for surgery *	Recurrence beyond two years after treatment (C-index = 0.73)	Liu, et al., 2020 [30]
HCC: Prediction of treatment outcome after TACE	130 cases *	Response for TACE AURUC = 0.93	Liu, et al., 2020 [90]
CT			
Differential diagnoses of HCC and non-HCC	211 cases (97 HCC and 124 non-HCC cases)	Accuracy = 0.824 (radiomic data from CT and MRI were applied)	Ma, et al., 2024 [81]
HCC: Prediction of early recurrence after hepatectomy	124 cases (with cirrhosis)	Model integrating radiomics and clinical–radiological characteristics AUROC of 0.844 in the training set, AUROC of 0.790 in the validation The Kaplan–Meier curves assessing OS revealed statistically significant differences.	Qian, et al., 2024 [83]
HCC (intermediate stage): Prediction of treatment response to TACE	367 cases	AUROC of 0.800 in external validation set The model stratifies patients into responders and non-responders with distinct survival (*p* = 0.001)	Zhang, et al., 2024 [89]
HCC (intermediate stage): Prediction of treatment outcome after TACE	543 cases	Time to progression after TACE (C index = 0.70)	Wang, et al., 2022 [107]
HCC: Prediction of treatment outcome after TACE	313 cases *	Response for TACE AUROC = 0.92	Peng, et al., 2022 [108]
HCC: Prediction of treatment outcome after TACE	111 cases *	Response for TACE AUROC = 0.91	Bai, et al., 2022 [109]
HCC: Prediction of treatment outcome after TACE	48 cases	Response for TACE AUROC = 0.90	Li, et al., 2022 [110]
HCC: Prediction of treatment outcome after TACE	248 cases *	Response for TACE AUROC = 0.87	Li, et al., 2022 [111]
HCC: Prediction of survival and risk of hepatic decompensation after TACE using splenic volume (SV)	327 cases	Survival was significantly lower in patients with high SV compared to low SV (*p* = 0.001). Patients with a hepatic decompensation after TACE had significantly higher SV (*p* < 0.001).	Muller, et al., 2022 [92]
HCC: Prediction of microvascular invasion	405 cases	AUROC = 0.980 in training set and 0.906 in validation set	Jiang, et al., 2021 [77]
HCC: Prediction of recurrence after liver transplantation	88 cases	Tumor recurrence/progression after transplantation AUROC = 0.87	Ivanics, et al., 2021 [112]
HCC (intermediate stage): Prediction of treatment outcome after TACE	310 cases	Response for TACE AUROC = 0.99	Peng, et al., 2021 [113]
HCC: Prediction for TACE ineligibility	256 cases *	Emergence of extrahepatic metastasis and vascular invasion after TACE. AUROC = 0.91	Jin, et al., 2021 [114]
HCC: Prediction of treatment outcome after TACE	789 cases	Response with 4-class classification (CR, PR, SD, PD) Accuracy = 85.1%	Peng, et al., 2020 [86]
HCC: Prediction for TACE ineligibility	243 cases *	Response for TACE AUROC = 0.90	Liu, et al., 2020 [87]
HCC: Prediction of treatment outcome after TACE	105 cases *	Response for TACE Accuracy = 0.742	Morshid, et al., 2019 [85]
Prediction of radiation-induced liver injury	125 cases (including 36 HCC cases)	Emergence of radiation-induced liver injury AUROC = 0.85	Ibragimov, et al., 2018 [91]
MRI			
HCC: PFS after MWA	149 cases *	Two-year RFS (C-index = 0.73)	Peng, et al., 2023 [115]
HCC: Prediction of treatment outcome after TACE	140 cases *	Response for TACE AUROC = 0.81	Liu, et al., 2022 [116]
HCC: Prediction of treatment outcome after TACE	252 lesions	Response for TACE (three-class classification, accuracy = 93.2%)	Svecic, et al.,2021 [117]
HCC: Prediction of microvascular invasion	158 cases for training (79 cases for validation)	AUROC = 0.81 (sensitivity = 69%, specificity = 79%) in training set, AUROC = 0.72 (sensitivity = 55%, specificity = 81%) in validation set	Zhang, et al., 2021 [78]
HCC (solitary, 2~5 cm in size): RFS after surgery	167 cases	Model trained with images of the 3 mm peritumoral border extension of tumor showed comparable performance to that of the postoperative clinicopathological model	Kim, et al., 2019 [80]
HCC: Prediction of microvascular invasion	110 cases	Presence of microvascular invasion sensitivity = 0.90, specificity = 0.75, accuracy = 0.83	Feng, et al., 2019 [79]
HCC: Prediction of treatment outcome after TACE	36 cases *	Response for TACE accuracy = 78%, sensitivity = 62.5%, specificity = 82.1%	Abajian, et al., 2018 [84]
Pathology			
Pathological diagnosis of HCC	HE-stained WSIs from 592 HCC cases	87.81% pixel-level accuracy and 98.77% slide-level accuracy in the test set 87.90% accuracy in validation set (from the Cancer Genome Atlas dataset)	Feng, et al., 2021 [97]
HCC: Prediction of survival after surgery	The Zhongshan cohort, 2451 images The TCGA cohort, 320 images (whole-slide image) and multiomics data	A ‘tumor risk score (TRS)’ was established to evaluate patient outcomes. The predictive ability of TRS was superior to and independent of clinical staging systems	Shi, et al., 2021 [103]
HCC: benign and malignant classification, differentiation, prediction of mutation	Histopathological HE images from the Genomic Data Commons Databases	96.0% accuracy for benign and malignant classifications 89.6% accuracy for well, moderate, and poor tumor differentiation Mutation of five major kinds of genes could be predicted (AUROCs from 0.71 to 0.89)	Chen, et al., 2020 [95]
Differentiation between HCC and CC on HE WSI	70 (35 HCCs and 35 CCs) digital WSIs 25,000 non-overlapping image patches	Accuracies of 0.885 on a validation set and 0.842 on an independent test set	Kiani, et al., 2020 [96]
HCC: Prediction of survival after surgery	The discovery set, 194 images The validation set, 328 images (WSIs)	C-index = 0.75~0.78	Saillard, et al., 2020 [102]

* Report of the models using image data along with clinical and blood test data for training. HCC: hepatocellular carcinoma, RFS: recurrence-free survival, DL: deep learning, MWA: microwave ablation, RFA: radiofrequency ablation, machine learning, TACE: transarterial chemoembolization, CEUS: contrast-enhanced ultrasound AUROC: area under the receiver operating characteristic curve, CR: complete response, PR: partial response, SD: stable disease, PD: progressive disease, TCGA: the Cancer Genome Atlas, CC: cholangiocarcinoma, WSIs: whole-slide images.

## 6. Clinical Significance

Many studies underscore the vital role of AI-enhanced imaging in non-invasive prognostic predictions for liver disease, offering significant potential for the advancement of personalized medicine. Currently, several AI-based systems that support the diagnosis of liver disease have been approved by the U.S. Food and Drug Administration (Table 3).

Integrating radiomic features with clinical and genetic insights into AI models improves the accuracy and decision-making speed of diagnosis and treatment in clinical settings. However, despite these promising applications, the deployment of AI in clinical practice presents multiple challenges. A model’s performance largely depends on high-quality, diverse datasets, yet many studies suffer from small sample sizes, incomplete data, and inconsistencies across datasets. Furthermore, AI models’ “black-box” nature limits their transparency and interpretability, which is crucial for gaining clinicians’ trust. Therefore, improved model explainability and interpretability are essential to facilitate the integration of AI into the clinical workflow. Additionally, the regulatory landscape remains complex, with ongoing efforts being made by the Food and Drug Administration and the European Commission to establish guidelines for the use of AI in healthcare, highlighting the need for safety and intellectual property considerations as AI technologies evolve. Therefore, for broader clinical adoption, further efforts are necessary in the following areas: (1) enhancing the generalizability of AI algorithms across diverse clinical environments, (2) conducting direct comparative trials between AI and conventional diagnostic methods, and (3) validating the performance of AI in clinical practice. Specifically, algorithms need to be optimized for different populations and clinical settings, prospective clinical trials should be required to confirm the superiority of AI over traditional methods, and large-cohort studies are essential for ensuring the reliability of AI in real-world clinical applications.

## 7. Conclusions

The application of AI in liver imaging diagnostics is promising, but still in its early stages. Challenges remain, particularly in ensuring the quality of image data and the accuracy of ground truth labels. The effectiveness of AI models relies heavily on the quality and volume of training data, making it essential to collect large, high-quality, and accurately labeled datasets to support further AI development. Additionally, AI models that integrate multimodal data, combining medical imaging with pathological, genomic, epigenomic, and transcriptomic information, show great potential for advancing personalized medicine [7,21,118]. With continued development, AI is poised to revolutionize the diagnosis and treatment of liver diseases.

## Figures and Tables

**Figure 1 bioengineering-11-01243-f001:**
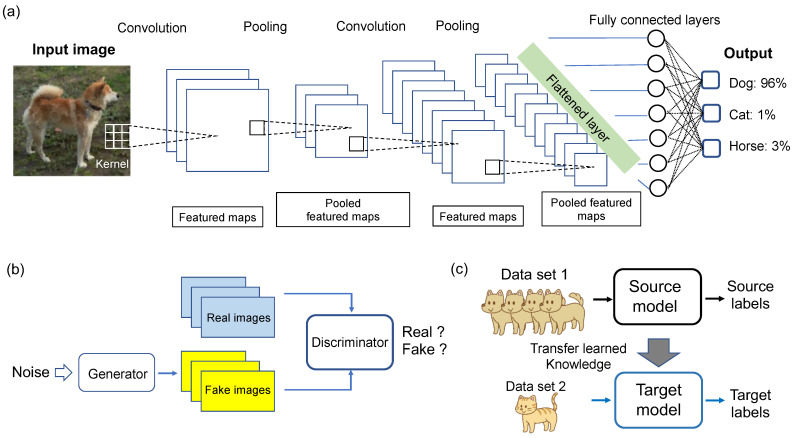
The overview of AI models commonly used for training with image data. (**a**) Convolutional neural networks (CNNs) are specialized AI frameworks designed to process grid-like data structures such as images. They comprise convolutional layers to detect features through filters (kernels), pooling layers that reduce dimensionality and enhance computational efficiency, with non-linear activation functions to capture complex relationships, and fully connected layers for generating predictions. By combining these components, CNNs learn hierarchical representations and complex patterns. (**b**) Generative adversarial networks (GANs) are composed of two neural networks based on unsupervised (label-free) learning. The generator creates fake data by mapping random noise, which serves as the seed for the features of the generated data, to resemble the target data. The other network, the discriminator, is tasked with distinguishing between the fake data generated by the generator and real data, determining their authenticity. (**c**) Transfer learning applies knowledge from one task to another, enabling effective learning even with limited data. For example, when building classifiers for dogs and cats, a large dog image dataset may suffice for training, but a small cat dataset may limit accuracy. Transfer learning leverages a pre-trained model from a related domain, replacing and training only the output layer to adapt it to the new task. This approach enhances model accuracy even with minimal data and reduces the time for training.

**Table 1 bioengineering-11-01243-t001:** Stage of medical care and AI application required in each step.

Stage	Prevention	Diagnosis	Treatment
Actions required	Improvement of lifestyle	Narrowing down high-risk populations Examination with high accuracy	Selection of effective treatments
AI support	Encouraging behavior and lifestyle change	Screening assistance and diagnostic support	Prediction of treatment outcomes

**Table 3 bioengineering-11-01243-t003:** FDA-approved AI-assisted imaging solutions applicable for supporting the diagnosis of liver disease.

Product	Function	Application	Vender
Velacur	AI-guided 3D share-wave US elastography device that measures the key indicators of SLD: liver stiffness, attenuation, and fat fraction with accuracy comparable to MR elastography.	Early diagnosis of SLD, provides actionable data to clinician without MRI settings.	SonicIncytes Vancouver, Canada
Liver Fat Quantification	A tool for the quantification of liver fat on US.	Early detection and management of SLD.	Philips Amsterdam, Netherlands
POCUS Pro	A web-based real-time guidance on liver US examination.	To improve accuracy in capturing high-quality images.	Clarius Vancouver, Canada
VisAble.IO	AI for planning liver tumor ablation therapy, providing 3D visualization of the ablation zones in real time.	Ablation of liver tumor with accurate ablation margins and minimally invasive procedures.	TechsoMed Rehovot, Israel
BioTraceIO	This system complements VisAble.IO.	A comprehensive image-guided ablation visualization framework, eliminating the guesswork.	TechsoMed Rehovot, Israel
LiverMultiScan	MRI-based software for the assessment of liver tissue characteristics such as the deposition of fat and iron content.	Diagnosis of SLD and fibrosis without liver biopsy.	Perspectum Oxford, UK
Quantib AI for MRI	AI that helps identify liver lesions and provide quantitative analysis on MRI images.	Diagnosis of liver tumors and monitoring.	Quantib Rotterdam, Netherlands
Advanced Intelligent Clear-IQ Engine (AiCE)	AI-based technology for CT and MRI that helps enhance image clarity, making it easier to identify liver lesions.	Easy identification of liver lesions and other abnormalities.	Canon Tokyo, Japan
SubtleMR	AI for the improvement of MRI imaging, reducing noise and enhancing the quality.	To help radiologists better detect liver lesions, particularly in the early stages.	SUBTLR MRDICAL Menlo Park, California
HEPATIQ^®^	Using nuclear medicine images from a SPECT scanner, this system computes the PHM^®^ liver function index, fLV^®^ liver volume index, fSV^®^ spleen volume index, HAI™ activity index, eFS™ fibrosis score, and eEV™ variceal size.	Determination of the liver’s functional reserve for the management of liver disease.	Hepatiq, Inc Irvine, California
Aidoc’s AI solutions	AI for multiple imaging modalities, including liver analysis.	To provide real-time alerts for radiologists identifying liver lesions and abnormalities.	Aidoc Tel Aviv, Israel
Curie|ENDEX™	AI that transform imaging data to standardized nomenclature and enables relevant clinical content linkage across disparate systems.	Assisting in liver cancer diagnosis, staging, and monitoring.	Enlitic Fort Collins, Colorado

AI: artificial intelligence, SLD: Steatotic liver disease, US: ultrasound.

## Data Availability

Data in this review paper are openly available from references.

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
