# Peer review of "Advancements in Artificial Intelligence-Enhanced Imaging Diagnostics for the Management of Liver Disease—Applications and Challenges in Personalized Care"

_bioengineering, 2024, doi:10.3390/bioengineering11121243_

Round 1

Reviewer 1 Report

Comments and Suggestions for Authors

1. The novelty of the review article is missing.

2. Very few studies are considered for the work.

3. The title says that advancement in imaging and medical AI ... but not discussed the medical imaging in entire manuscript.

4. The medical AI terminology is not defined and discussed in manuscript. If author assumed that the medical AI and AI based diagnosis are same then author has to cite the suitable reference.

5. Language correction required.

6. The entire manuscript seems like a short notes. So author has to rewrite the entire manuscript in systematic manner.

7. limited references are considered. Author has to explore more about the domain.

Comments on the Quality of English Language

Minor correction required.

Author Response

Dear reviewer

I sincerely appreciate your evaluation of my manuscript and the insightful comments you provided. I would like to respond as follows:"

Comment 1

  1. The novelty of the review article is missing.

Reply 1

Thank you for your important comments. The primary objective of this manuscript is to summarize the latest advancements in AI-supported imaging diagnosis for liver diseases, organize existing research findings, and present the current state, challenges, and future perspectives of this field. Notably, this review focuses on the application of AI in imaging diagnosis for personalized medicine in liver diseases, which we believe represents a unique perspective. These insights are specifically addressed in the sections “4. AI-aided imaging diagnosis and its clinical application” and “5. Prognostic prediction and personalized medicine in liver diseases with imaging-based AI.”

Comment 2

  1. Very few studies are considered for the work.

Replay 2.

In response to the reviewers' comments, we have incorporated a discussion of 41 new papers published since 2022 into the main text, thereby enhancing the content of the manuscript.

Comment 3

  1. The title says that advancement in imaging and medical AI ... but not discussed the medical imaging in entire manuscript.

Replay 3

The manuscript presents advancements in AI-enhanced diagnostic support technologies in the field of liver disease. In accordance with the reviewers' suggestions, the title has been revised to "Advancements in AI-enhanced Imaging Diagnostics for the Management of Liver Disease" by removing the term "medical imaging".

Comment 4

  1. The medical AI terminology is not defined and discussed in manuscript. If author assumed that the medical AI and AI based diagnosis are same then author has to cite the suitable reference.

Replay 4

This manuscript introduces AI-based imaging diagnostic support technologies in the field of liver disease. In response to the reviewers' comments, the term "medical AI" has been removed from the text and replaced with "AI-enhanced imaging diagnostics" or "AI-aided imaging diagnosis.

Comment 5

  1. Language correction required.

Replay 5

The manuscript has been submitted for further editing to Editage (www.editage.com) for paid English language proofreading and refinement of the expressions.

Comment 6

  1. The entire manuscript seems like a short notes. So author has to rewrite the entire manuscript in systematic manner.

Replay 6

In response to the reviewers' comments, the manuscript has been significantly revised and enhanced. The changes are highlighted in red within the text. Notably, the central section titled "AI-aided imaging diagnosis and its clinical application" has been expanded. The subsection "4.1. Prediction of staging and diagnosis of lesions" has been restructured to include "4.1. AI-assisted imaging in the diagnosis of hepatic steatosis and fibrosis," with the discussion on fibrosis separated into "4.1.1. AI models for staging and early diagnosis of liver fibrosis" and the section on steatosis into "4.1.2. AI models for staging and early diagnosis of steatotic liver disease."

Additionally, a new subsection titled "4.2. AI Models for Diagnosis of Liver Tumor" has been added to address AI applications in imaging diagnosis of liver tumors. The sections "4.2. Risk prediction of liver disease" and "4.3 Application to personalized medicine for the management of liver cancer" have been reorganized under a new heading, "5. Prognostic prediction and personalized medicine in liver diseases with imaging-based AI." Within this section, the discussion on AI applications related to the management of complications in liver cirrhosis has been specified as "5.1. Applications of AI-assisted imaging in management of the complications in liver cirrhosis," while the management of liver cancer has been detailed in "5.2 AI-assisted imaging in diagnosis and management of HCC."

Furthermore, a new section titled "6. Clinical Significance" has been created to introduce AI-based technologies currently approved by the FDA, discussing their challenges and future prospects.

Comment 7

  1. Limited references are considered. Author has to explore more about the domain.

Replay 7

In accordance with the reviewers' suggestions, the manuscript has been enhanced by incorporating the findings of 41 new studies published since 2022, significantly enriching the content of each section.

  • Regarding the "1. Introduction" section: A discussion of the current challenges in liver disease management and the introduction of AI-based solutions has been added on pages 1-2, lines 26-45.
  • Regarding the "2. Advantages of AI in medical imaging diagnostics" section: Detailed information on the advantages of using AI for imaging diagnostics in liver disease has been included on pages 2, lines 71-85.
  • Regarding the "3. Training of AI models using imaging data of liver disease" section: A conceptual diagram of the AI algorithms has been presented in Figure 1 to aid understanding.
  • Regarding the "4. AI-aided imaging diagnosis and its clinical application" section: An introductory description for this section has been added on pages 6, lines 230-239. In subsection "4.1.1. AI models for staging and early diagnosis of liver fibrosis," a detailed introduction to AI predicting liver fibrosis has been provided on pages 6-7, lines 240-281. Similarly, in subsection "4.1.2. AI models for staging and early diagnosis of steatotic liver disease," an introduction to AI predicting hepatic steatosis has been included on pages 7-8, lines 282-330. Additionally, a new subsection titled "4.2. AI Models for Diagnosis of Liver Tumor" has been added to discuss AI applications in imaging diagnosis of liver tumors (pages 8-9, lines 331-391).
  • Regarding the "5. Prognostic prediction and personalized medicine in liver diseases with imaging-based AI" section: In subsection "5.1. Applications of AI-assisted imaging in management of the complications in liver cirrhosis," a detailed introduction to AI supporting the management of complications related to liver cirrhosis has been added (pages 9, lines 392-404; page 10, lines 415-458). Furthermore, in subsection "5.2 AI-assisted imaging in diagnosis and management of HCC," new research findings have been included (page 10, line 459 - page 11, line 482; page 11, lines 490-497; page 11, lines 504-512; page 12, lines 530-548; page 12, lines 554-575).

Additionally, ten new studies have been included in Table 2 to present their findings.

  • Regarding the "6. Clinical Significance" section: An introduction to AI technologies related to imaging diagnostics for liver diseases that have been approved by the FDA has been added (pages 16, lines 582-609, and Table 3).

Reviewer 2 Report

Comments and Suggestions for Authors

Author aimed to outline the current state of AI development in medical imaging for liver diseases 

and its societal implementation, focusing primarily on deep learning models. 

This is an interesing paper.

There are several minor comments.

1) Please, provide the schematic figure to understand the AI methods (e.g. CNN, GANs..)

2) It would be better to add the section regarding the AI-based differential diagnosis of liver nodule (FNH, adenoma, HCC, cholangiocarcinoma, and combiend HCC-cholangiocarcinoma).

3) Please, mention commercially available AI-based imaging tools, if any.

Author Response

Dear reviewer

I sincerely appreciate your evaluation of my manuscript and the insightful comments you provided. I would like to respond as follows:"

Comments and Suggestions for Authors

Author aimed to outline the current state of AI development in medical imaging for liver diseases and its societal implementation, focusing primarily on deep learning models. 

This is an interesting paper.

Replay

I sincerely appreciate your evaluation of my manuscript and positive comments for it.

Comment 1

1) Please, provide the schematic figure to understand the AI methods (e.g. CNN, GANs..)

Replay 1

We appreciate the valuable and important comments provided. A figure 1 has been added on page 4 to present a conceptual diagram of the AI algorithms, which aims to enhance understanding.

Comment 2

2) It would be better to add the section regarding the AI-based differential diagnosis of liver nodule (FNH, adenoma, HCC, cholangiocarcinoma, and combiend HCC-cholangiocarcinoma).

Replay 2

A new section has been added under the subheading "4.2. AI Models for Diagnosis of Liver Tumor" to discuss the role of AI in supporting imaging diagnosis for the differential diagnosis of liver tumors. This section elaborates on studies related to the differentiation of focal FNH, HCC, cholangiocarcinoma, and special types of HCC, as well as the associated challenges (pages 8-9, lines 331-391).

Comment 3

3) Please, mention commercially available AI-based imaging tools, if any.

Replay 3

A new section titled "6. Clinical Significance" has been added to introduce AI applications related to imaging diagnosis of liver diseases that have received FDA approval. This information has been summarized in Table 3 (pages 16, line 582 - page 17, line 609, and Table 3).